A metagenomic insight into freshwater methane-utilizing communities and evidence for cooperation between the Methylococcaceae and the Methylophilaceae

Beck David A.C. 1
Kalyuzhnaya Marina G. 2
Malfatti Stephanie 3 4
Tringe Susannah G. 4
Glavina del Rio Tijana 4
Ivanova Natalia 4
Lidstrom Mary E. 5
Chistoserdova Ludmila milachis@uw.edu 6
1 Department of Chemical Engineering and eScience Institute, University of Washington , Seattle, WA , USA
2 Department of Microbiology, University of Washington , Seattle, WA , USA
3 Lawrence Livermore National Laboratory , Livermore, CA , USA
4 DOE Joint Genome Institute , Walnut Creek, CA , USA
5 Departments of Chemical Engineering and Microbiology, University of Washington , Seattle, WA , USA
6 Department of Chemical Engineering, University of Washington , Seattle, WA , USA
Souza Valeria
Electronic publication date: 2013 Feb 19
Publication date: 2013
Volume: 1
Electronic Location ID: e23
Received 2012 Dec 1; Accepted 2013 Jan 9
Copyright: © 2013 Beck et al.
Copyright year: 2013
Copyright holder: Beck et al.
License: This is an open access article distributed under the terms of the Creative Commons Attribution License, which permits unrestricted use, distribution, and reproduction in any medium, provided the original author and source are credited.
License URL: https://creativecommons.org/licenses/by/3.0/

Keywords: Methane, Nitrate, Methylotrophy, Methylococcaceae, Methylophilaceae, Metagenomics, Microbial community, Lake sediment

Funding: National Science Foundation MCB-0604269 MCB-0950183 Department of Energy DE-SC0005154 This research was supported by the National Science Foundation (grants MCB-0604269 and MCB-0950183) and the Department of Energy (grant DE-SC0005154). This work was facilitated through the use of advanced computational, storage, and networking infrastructure provided by the Hyak supercomputer system, supported in part by the University of Washington eScience Institute. The work conducted by the US Department of Energy Joint Genome Institute was supported by the Office of Science of the US Department of Energy under contract no. DE-AC02-05CH11231. The funders had no role in study design, data collection and analysis, decision to publish, or preparation of the manuscript.

==============================
We investigated microbial communities active in methane oxidation in lake sediment at different oxygen tensions and their response to the addition of nitrate, via stable isotope probing combined with deep metagenomic sequencing. Communities from a total of four manipulated microcosms were analyzed, supplied with 13C-methane in, respectively, ambient air, ambient air with the addition of nitrate, nitrogen atmosphere and nitrogen atmosphere with the addition of nitrate, and these were compared to the community from an unamended sediment sample. We found that the major group involved in methane oxidation in both aerobic and microaerobic conditions were members of the family Methylococcaceae, dominated by species of the genus Methylobacter, and these were stimulated by nitrate in aerobic but not microaerobic conditions. In aerobic conditions, we also noted a pronounced response to both methane and nitrate by members of the family Methylophilaceae that are non-methane-oxidizing methylotrophs, and predominantly by the members of the genus Methylotenera. The relevant abundances of the Methylococcaceae and the Methylophilaceae and their coordinated response to methane and nitrate suggest that these species may be engaged in cooperative behavior, the nature of which remains unknown.

Introduction

Methane is recognized as one of the most powerful greenhouse gases, with annual emissions of approximately 600 Tg (King, 1992; Hanson & Hanson, 1996; Etiope & Klusman, 2002; Keppler et al., 2006). Its atmospheric concentration has been steadily increasing over the past 300 years, mostly due to anthropogenic activities (Singh et al., 2010). Until recently, two major modes have been recognized by which methane is removed from the environment: aerobic oxidation conducted by a specialized group of bacteria, known as methanotrophs (Hanson & Hanson, 1996; Chistoserdova & Lidstrom, 2013), and anaerobic oxidation linked to sulfate reduction, conducted by a specialized group of archaea, known as anaerobic methanotrophs or ANME (Valentine, 2002; Knittel & Boetius, 2009). The former process is important for methane consumption in freshwater sediments and soils, whereas the latter is thought to be the major process in anoxic marine environments. More recently, however, evidence has been accumulating that other metabolic modes for methane consumption must exist, linked to alternative electron donors, such as nitrate/nitrite-dependent anaerobic/microaerobic bacterial methane oxidation in freshwater environments (Wu et al., 2011) and metal-dependent methane oxidation by archaea in marine environments (Beal, House & Orphan, 2009). These new findings point toward novel biogeochemical processes that need elucidation in order to be placed into the context of the global carbon cycle. However, the relative environmental significance of these processes, the identity of the microbes involved, and the details of their metabolism remain poorly characterized. On the other hand, the clear separation between the aerobic and the anaerobic modes of metabolism may represent an artifact originating from experimental data predating environmental microbiology approaches, including culture-independent approaches. This notion is nicely illustrated by anaerobic nitrite-dependent methane oxidation in members of the NC10 phylum occurring by aerobic methane oxidation. This metabolic mode involves canonical methane monooxygenase, the classic sets of methanol and formaldehyde oxidation enzymes, and a strict reliance on the presence of oxygen that, in this case, is produced intracellularly (Wu et al., 2011). At the same time data are available suggesting that at least some of the classic aerobic methanotroph and methylotroph species may be able to thrive in microaerobic environments and potentially utilize alternative electron acceptors, such as nitrate, for methylotrophic metabolism (Costa et al., 2000; Modin, Fukushi & Yamamoto, 2007; Kalyuzhnaya et al., 2009; Stein & Klotz, 2011).

We have previously characterized communities involved in methylotrophy in Lake Washington, Seattle, USA, using both culture-reliant and culture-independent approaches, focusing on organisms active in aerobic conditions. These studies identified a diverse functional community and highlighted the potential importance of the Methylococcaceae and the Methylophilaceae species as members of this community (Nercessian et al., 2005; Kalyuzhnaya et al., 2008; Chistoserdova, 2011a). Metagenome-based metabolic reconstruction of these species has indicated that at least some of them are capable of denitrification, suggesting that they may be adaptable to an anaerobic/microaerobic life style (Kalyuzhnaya et al., 2008; Kalyuzhnaya et al., 2009). In this study, we have expanded the previous efforts of characterizing functional methylotroph communities by addressing the nature of the communities involved in methane metabolism in both aerobic and microaerobic conditions. In addition, we have addressed the potential role of nitrate in these communities in an attempt to further link carbon and nitrogen cycles in terrestrial environments.

Materials and Methods

Experimental setup, sample collection and stable isotope probing

The schematic of the experiments conducted is depicted in Fig. 1. Sediment samples were collected on March 3, 2009, from a 63 m deep station in Lake Washington, Seattle, Washington (47.038075’ N, 122.015993’ W) using a box core that allowed collection of undisturbed sediment. Samples were transported to the laboratory on ice and immediately used to set up microcosms. In order to assess populations active in methane oxidation under different oxygen tensions and to test for their potential dependence on the presence of nitrate we set up microcosm incubations as follows. One microcosm was incubated in an atmosphere of 50% 13C-labeled methane (99 atom % 13C, Sigma-Aldrich) and 50% ambient air, to assess the populations active in aerobic methane oxidation (the +O2-NO3- condition); the second microcosm was incubated in an atmosphere of 50% 13C-methane and 50% ambient air, in the presence of 10 mM KNO3, to assess the populations active in aerobic methane oxidation positively responding to the presence of nitrate (the +O2+NO3- condition); the third microcosm was incubated in an atmosphere of 50% 13C-methane and 50% N2, to assess the populations active in microaerobic methane oxidation (the -O2-NO3- condition); the fourth microcosm was incubated in an atmosphere of 50% 13C-methane and 50% N2, in the presence of 10 mM KNO3, to assess the populations active in microaerobic methane oxidation positively responding to the presence of nitrate (the -O2+NO3- condition). Each microcosm contained 50 ml of the top layer (1 cm) of the sediment and 50 ml of Lake Washington water filtered through 0.22 µm filters (Millipore). Samples were placed into 250 ml glass vials (6 vials per experiment, the contents of which were mixed before DNA extraction) and these were sealed with rubber stoppers. The duration of the incubation time for each microcosm was determined empirically by observing the formation of a heavy, 13C-enriched DNA fraction. It took 10 days for heavy DNA band to appear in the +O2+NO3- microcosm, compared to 15 days for the +O2-NO3- microcosm, suggesting that the methane-consuming community was stimulated by nitrate. It took much longer for heavy DNA band to appear in the microaerobic microcosms (20 and 30 days, respectively, for nitrate-amended and nitrate-free conditions; Fig. 1). These data suggest that nitrate also had a positive effect on methane consumption by the microbial community in microaerobic conditions.

Figure 1 Schematic of experimental setup shows workflow, duration of each enrichment and actual DNA samples separated into heavy and light fractions.

Community DNA was extracted as described previously (Beck et al., 2011) with one modification as follows: DNA samples were subjected to an additional round of purification using UltraClean® Mega Soil DNA Isolation Kit (MOBIO). The heavy (13C-enriched) fractions of DNA were separated from the light (12C) fractions by CsCl-ethidium bromide density gradient ultracentrifugation, visualized under UV (Fig. 1), and collected and purified following standard procedures as previously described (Neufeld et al., 2007). DNA from an untreated sediment sample was extracted using the same protocol. This sample was not subjected to density gradient centrifugation.

DNA sequencing and assembly

Five shotgun libraries were constructed, one from each microcosm. The libraries were sequenced using the 454 sequencing technology at the Joint Genome Institute (JGI) Production Genomics Facility. A total of 5,241,266 reads comprising 1.67 gigabases (Gb) of sequence were generated. These were assembled using the Newbler assembler. Assembly statistics are shown in Table 1.

Table 1 Sequencing, assembly, metagenome and pyrotag statistics.

	Unamended	+O2-NO3-	+O2+NO3-	-O2-NO3-	-O2+NO3-	
Number of raw metagenome reads (median read length)	1,644,561 (415)	1,601,297 (469)	342,227 (435)	206,554 (332)	1,446,627 (457)	
Size (base pairs)	559,537,102	308,706,277	80,246,742	59,711,589	354,011,745	
DNA scaffolds	1,515,849	835,955	193,120	194,103	925,371	
Number of contigs in the assembly	9,658	19,257	8,470	573	4,087	
Number of bp in assembled contigs	7,058,808	12,145,462	7,540,157	326,365	2,187,627	
N50 contig length, bp	985	803	1,459	935	802	
Mean coverage of assembled contigs	3.9	3.4	5.7	5.1	3.8	
Genes	1,554,721	821,124	216,380	160,657	948,029	
Proteins	1,547,567	817,673	215,668	159,899	943,870	
RNA genes	7,154	3,451	712	758	4,159	
16S rRNA genes	488	211	22	59	273	
16S rRNA genes curated (methylotroph genes)	458 (19)	186 (49)	22 (7)	50 (2)	261 (11)	
Raw pyrotag sequences (median read length)	21,348 (471)	6,457 (403)	27,720 (479)	16,364 (367)	27,714 (475)	
Pyrotag gene clusters	1,486	313	709	561	1,386	
Pyrotag diversity index (Shannon–Weaver)	4.10	3.84	2.40	3.94	4.01	
COG clusters	4,494	4,246	3,639	3,652	4,313	
Pfam clusters	5,788	5,199	3,822	3,861	5,157	
pmoB genes	31	106	30	0	9	
fae genes	62	157	38	3	13	
Nitrate reductase genes	902	634	224	44	523	

Pyrotag sequencing

We used 454 pyrotag sequencing of the V8 hypervariable region of the 16S rRNA gene to determine the compositions of microbial communities in the four 13C-enriched metagenomes using a computational pipeline PyroTagger, as previously described (Kunin & Hugenholtz, 2010), and we compared these to the community composition in the original (unamended) lake sediment sample. The number of total pyrotag sequences generated per microcosm is shown in Table 1. The PyroTagger assignments were used to calculate the Shannon-Weaver diversity index and the expected number of unique species rarefied to a constant value across all samples (Table 1).

Metagenome analysis

The draft quality assemblies were processed using the IMG/M pipeline (Markowitz et al., 2012) and the outputs of these automated analyses were manually verified and used for metagenome profiling.

Single gene taxonomy

To classify the 16S rRNA gene sequences in the metagenomes, the 16S genes identified by JGI pipeline were aligned against the Ribosomal Database Project (RDP) version 10, release 26. The top scoring alignment for each sequence was used to assign taxonomy, based on the annotations in the RDP. Genes were classified to the family level.

To detect pmoB and mmoX genes, respective peptides representing Methylococcaceae and Methylocystaceae were obtained from public databases and used as queries in pBLAST analyses against each dataset, and all matches were recorded. These queries pick each other and also the NC10 sequences. All matches were compared with sequences in public databases using pBLAST. Genus level assignments were made using an 80% identity cutoff level, and family level assignments were made at 70% identity cutoff, based on prior knowledge on gene diversity within each family (Chistoserdova & Lidstrom, 2013).

To detect fae genes, respective peptides representing Methylococcaceae, Methylocystaceae, and Methylophilaceae were obtained from public databases and used as queries in pBLAST analyses against each dataset, and all matches were recorded. These queries pick each other, other proteobacterial sequences as well as NC10, planctomycete and unclassified sequences. Genus and family level assignments were done as above.

For the nitrogen metabolism genes, sequences were selected by their annotation and aligned against the NCBI’s nonredundant database (nt). The top scoring hit for each sequence was saved. The NCBI taxon ID was extracted and the NCBI taxonomy database queried to collect the “scientific name” of each hit. The count of hits for each “scientific name” were reported.  The following annotation search terms were used: “itrate reductase” for NO3-reductase, “itrite reductase” for NO2-reductase, “itric[-]oxide reductase” for NO reductase, “itrous[-]oxide reductase for N2O reductase, and “itrogenase” for nitrogenase. The first letter was omitted to avoid conflicts with upper and lower case letters and in the case of NO and N2O reductases, both annotations (with and without “-”) were accepted.

To classify mxaF and xoxF genes belonging to specific methylotroph families, high stringency pBLAST analyses were applied using 90% identity level cutoff at the protein level and employing publically available sequences of the respective enzymes representing the respective taxonomic groups.

Figure 2 Taxonomic profiling of microcosms based on pyrotag analysis (A and B) and metagenome data analysis (C and D) shows high abundance of Methylococcaceae and Methylophilaceae in aerobic conditions. A. Distribution of pyrotag sequences among major phyla. Other, phyla making up less than 1% of total. B. Proportions of Methylococcaceae and Methylophilaceae sequences in pyrotag libraries. C. Distribution of sequences in metagenomes taxonomically classified at 90% identity level. D. Proportions of Methylococcaceae, Methylophilaceae and Methylocystaceae of total sequences taxonomically profiled at 90% identity level.

Results

Pyrotag profiling of community DNA shows enrichment for Methylococcaceae and Methylophilaceae sequences in aerobic microcosms

As expected from prior analyses (Kalyuzhnaya et al., 2008), the community in the unamended sample revealed high complexity, being represented by a total of 1,486 sequence clusters (97% sequence identity; Kunin & Hugenholtz, 2010). The community was dominated by Proteobacteria (33.3%), of which phylotypes of the Methylococcaceae family that represents one class (called type I) of methane oxidizing bacteria were most prominently present (10% of all sequences). The second most dominant group was represented by chloroplast sequences (21.9%). Other prominently present phyla were Bacteroidetes (10.5%), Acidobacteria (7.2%) and Chloroflexi (4.0%; Fig. 2A). The phylogenetic complexity of the 13C-enriched metagenomes representing three of the enrichment conditions (the +O2-NO3- condition, the +O2+NO3- condition, and the -O2-NO3- condition) was significantly reduced compared to the non-enriched community (313, 709 and 561 sequence clusters, respectively), and different shifts in phyla distribution occurred in these communities. The proportion of proteobacterial sequences increased in the aerobic communities (to 49.7% and to 82.0%, respectively, in +O2-NO3- and in +O2+NO3- conditions; Fig. 2A). In both cases phylotypes classified as the Methylococcaceae and the Methylophilaceae were most prominently present (10.9% and 14.9% of all sequences in the +O2-NO3- condition and 56.5% and 10.5% of all sequences in the +O2+NO3- condition, respectively; Fig. 2B). The proportion of Proteobacteria decreased (to 13.2%) in the -O2-NO3- condition, the dominant phylotypes being the chloroplast and the Bacteroidetes phylotypes (34.0% and 9.5% of total sequences, respectively), while the Methylococcaceae and the Methylophilaceae phylotypes constituted only a minor fraction of all sequences (0.03% and 0.9%, respectively). The proportion of proteobacterial phylotypes and the overall make up of the community of the -O2+NO3- condition, at the phylum level, resembled that of the non-enriched community (Fig. 2B). However, the Methylococcaceae phylotypes were less represented (4.0% of total sequences) while other proteobacterial phylotypes (such as Burkholderiales) were more represented than in the non-enriched community. Phylotypes representing other bona fide methylotroph taxa, including members of the Methylocystaceae and Bejerinckiaceae that represent the second class (called type II) of methane oxidizers and members of the recently described NC10 phylum implicated in anaerobic methane oxidation linked to denitrification (Wu et al., 2011) were detected in all five pyrotag libraries. However, their proportions were very small, not exceeding 0.5% of the total community in each case.

Genome recruitment further highlights the dominant presence of Methylococcaceae and Methylophilaceae species in aerobic microcosms

A total of 1,362,213,455 base pairs (1.36 Gb) of assembled sequence were generated. Genes were called using the standard JGI IMG/M pipeline (Markowitz et al., 2012), and each gene was taxonomically classified by using its best BLAST hit in the current genomic database employed by the IMG/M interface (3811 bacterial genomes, 163 archaeal genomes, 177 eukaryotic genomes and 2803 viral genomes; sequencing and assembly statistics are shown in Table 1). For taxonomic assignments, we considered separately all protein coding genes and genes classified at the 60% and the 90% cutoff levels (protein level classification). While few genes were classified at the 90% cutoff level (2.44 to 17.56% of the total, dependent on the microcosm) these provide very robust proxies for the organisms represented in the metagenomes, especially given the fact that genomes of key model methylotrophs were parts of the database used for comparisons, including genomes originating from Lake Washington (Lapidus et al., 2011; Kittichotirat et al., 2011 and unpublished). Thus we mostly relied on the 90% cutoff classification for the confident sequence assignments, while realizing that these provide the lowest estimate for the presence of a specific phylum. At the 60% cutoff level, between 28.9% and 42.5% of the total genes could be classified, allowing for encompassing species not represented in the databases by very closely related models. In either case, most of the genes in each metagenome were taxonomically classified as proteobacterial, with the absolute majority matching to beta- and gammaproteobacteria (up to 45.4% and up to 50.1% of genes at 90% cutoff level, respectively). The next most abundant group were alphaproteobacteria (up to 12.4% of genes at 90% cutoff level), excepting the -O2-NO2- metagenome, in which, along with gamma- and betaproteobacteria, Bacteroidetes and deltaproteobacteria sequences were prominent. Remarkably, most of the gammaproteobacterial sequences (up to 84.2/98.4% at the 60/90% cutoff levels, dependent on the microcosm) were classified as belonging to the Methylococcaceae family. Among betaproteobacterial sequences, the largest proportion of sequences in the aerobic microcosms (up to 53.6/88.4%) were classified as belonging to the Methylophilaceae family, while in the microaerobic microcosms betaproteobacterial sequences were distributed among a number of dominant families, which included Methylophilaceae, Comamonadaceae, Rhodocyclaceae and Burkholderiaceae. Among the alphaproteobacterial sequences, those classified as belonging to the Methylocystaceae family only constituted the highest proportion (48.6/81.3%) in the +O2-NO3- microcosm, but they were also abundant in the +O2+NO3- microcosm (13.0/30.1% of total alphaproteobacterial sequences). In the remaining datasets, few alphaproteobacterial sequences were classified as Methylocystaceae, with Bradyrhizobiaceae sequences being most abundant. Similarly to the results of pyrotag analysis, sequences of other bona fide methylotrophs (such as Methylobacteriaceae, Bejerinckiaceae, Xanthobacteriaceae, Hyphomicrobiaceae) were identified in each metagenome. However, the proportions of the genes assigned to each of these families were low, suggesting that these species were minor members of the functional communities investigated.

Sequences assigned to each of the major methylotroph families, Methylococcaceae, Methylophilaceae and Methylocystaceae, were further classified at the genus level, by matching each gene classified to the family level to the available genomic scaffolds representing each family, at 60 and 90% cutoff values. In these analyses, the Methylococcaceae family was represented by a total of six genomes, of Methylobacter tundripaludum (Svenning et al., 2011), Methylomonas sp. (unpublished), Methylomicrobium album (unpublished), Methylocaldum szegediense (unpublished), Methylococcus capsulatus (Ward et al., 2004) and Crenothrix polyspora (unpublished). In the case of the latter, we find the claimed affiliation of this strain within a proposed new family of Crenothrichaceae (Stoecker et al., 2006) invalid, based on high sequence similarity with the members of Methylococcaceae (at least 95% at 16S rRNA gene level; (Iguchi, Yurimoto & Sakai, 2011), and thus we consider this organism as part of this family. However, the matters are complicated further by the fact that the sequence of the yet uncultivated C. polyspora originated from a highly enriched but not axenic culture that may contain other representatives of Methylococcaceae. Of the six genomes, only one represented an organism originating from Lake Washington, the Methylomonas sp. strain (unpublished data). The Methylocystaceae family was represented by only two genomes, of Methylosinus trichosporium (Stein et al., 2010) and Methylocystis sp. (Stein & Klotz, 2011), none originating from Lake Washington. The Methylophilaceae family was represented by eight genomes, of Methylotenera mobilis, Methylotenera versatilis, Methylovorus glucosotrophus (Lapidus et al., 2011), Methylophilus sp., unclassified Methylophilaceae (unpublished), Methylobacillus flagellatus (Chistoserdova et al., 2007), and unclassified Methylophilaceae strains HTCC8121 (Giovannoni et al., 2008) and KB13 (unpublished). Of these, the first five strains originated from Lake Washington (Kalyuzhnaya et al., 2006; Kalyuzhnaya et al., 2012 and unpublished) and the latter two were marine Methylophilaceae with unusually small genomes (Giovannoni et al., 2008). All the genomes mentioned here as “unpublished” have been sequenced by the JGI and are publically available through the IMG/M interface (http://img.jgi.doe.gov). From matching to this limited number of genomic scaffolds, the Methylobacter species appeared to be the dominant type among the Methylococcaceae representatives, the Methylocystis types appeared to dominate over the Methylosinus types, and Methylotenera species appeared to be dominant among the Methylophilaceae. However, specific dynamics could be observed at the genus level in response to different stimuli (Table 2). The proportion of the Methylomonas and Methylomicrobium types increased in response to the addition of methane, especially pronounced in the +O2+NO3- condition, while the proportion of the “Crenothrix” types decreased compared to unamended sediment. The Methylobacter types were most dominant in the -O2+NO3- condition. Dynamics were also clearly seen among the Methylocystaceae, with the proportion of Methylosinus sequences upshifting in response to methane in aerobic conditions and downshifting in response to nitrate, with respect to Methylocystis sequences. The relative proportion of Methylotenera sequences among the Methylophilaceae populations upshifted significantly in response to nitrate in aerobic conditions. We were also able to distinguish between two different species within Methylotenera, M. mobilis vs. M. versatilis, based on their significant divergence at the genomic level (Lapidus et al., 2011). While the proportion of M. versatilis-like sequences increased in the +O2+NO3-condition, their proportion decreased in the -O2+NO3- condition, being replaced by the M. mobilis-like sequences. Dynamics could also be seen among sequences classified as other Methylophilaceae. Unsurprisingly, virtually no sequences were matched to the scaffolds representing marine Methylophilaceae.

Table 2 Relative distribution of genes classified at 60%/90% cutoff among different genera. Sum of genes assigned to each family at each cutoff level equals 100%.

Genome	Unamended	+O2-NO3-	+O2+NO3-	-O2-NO3-	-O2+NO3-	
Methylococcaceae						
Methylobacter	47.7/60.6	50.9/64.0	47.2/64.8	35.0/58.1	36.4/74.3	
Crenothrix	30.0/30.1	19.4/17.5	12.7/5.2	38.5/31.0	36.2/19.0	
Methylomonas	10.8/4.3	14.6/10.2	20.2/19.8	10.9/6.2	10.4/3.4	
Methylomicrobium	7.4/3.8	11.4/7.5	17.6/10.1	9.1/4.0	8.0/2.9	
Methylocaldum	2.3/<1	2.3/<1	1.5/1.2	3.9/<1	5.6/<1	
Methylococcus	1.5/<1	1.4/<1	<1/<1	2.7/<1	3.4/<1	
Methylocystaceae						
Methylocystis	67.0/89.6	65.2/63.3	90.2/95.8	71.4/88.9	74.4/90.2	
Methylosinus	33.0/10.4	34.8/36.7	9.8/4.2	28.6/11.1	25.6/9.8	
Methylophilaceae						
Methylotenera versatilis	41.7/56.3	41.1/65.9	41.2/64.4	32.6/52.0	25.8/48.3	
Methylotenera mobilis	21.5/24.8	21.1/17.7	41.5/28.2	21.6/25.0	26.3/35.4	
Methylovorus	16.2/5.1	16.3/4.8	5.4/<1	17.8/9.7	19.4/3.3	
Methylobacillus	9.7/3.2	8.5/2.4	2.1/<1	12.7/4.6	13.7/2.9	
Methylophilus	3.4/2.3	2.9/1.8	2.4/<1	6.3/2.6	4.9/2.0	
Unclassified Methylophilaceae	7.0/7.9	9.7/7.3	7.3/5.1	8.5/6.1	8.7/7.8	
Marine Methylophilaceae	<1/<1	<1/<1	<1/<1	<1/0	1.2/<1	

Figure 3 Abundances of the Methylococcaceae and the Methylophilaceae sequences as per cent of total taxonomically classified sequences show good correlation.

Taxonomic profiling demonstrated a good correlation between the populations of the Methylococcaceae and the Methylophilaceae in both aerobic conditions (Figs. 2 and 3). Both were more abundant in the +O2+NO3- microcosm and somewhat less abundant in the +O2-NO3- microcosm, suggesting that both preferred higher nitrate concentrations for C1 metabolism. The low oxygen tension conditions selected against all methylotroph species. However, the Methylococcaceae still represented the majority of gammaproteobacterial sequences at the 90% cutoff level (Fig. 2D). Overall, taxonomic profiling of metagenomes correlated well with pyrotag-based profiling both suggesting that Methylococcaceae and Methylophilaceae efficiently consumed the 13C label from methane in aerobic conditions while the label distributed more evenly among multiple phyla in microaerobic conditions. Results of ordination analysis of dissimilarity of the five communities are shown in Supplemental Figure 1.

Single gene-based taxonomic profiling supports data from whole-metagenome profiling

16S rRNA gene profiling in each microcosm was carried out (Tables 1 and 3; Supplemental Table 1). For the metagenomes with significant sequence sampling (Table 1), the distribution of 16S rRNA genes among major phyla matched well those determined by the pyrotag sequencing approach, with some differences such as the reduced proportion of chloroplast sequences (data not shown), which is likely due to the low diversity of chloroplast sequences. Analysis of 16S rRNA gene sequences revealed that only in the aerobic microcosms did methylotroph sequences make up a significant fraction of total 16S rRNA gene sequences (26.3 to 31.8%, respectively, in the +O2-NO3- and the +O2+NO3- conditions; Table 1). In the microaerobic conditions and in the unamended sample, the methylotroph 16S rRNA gene fraction made up approximately 4% of the total 16S rRNA sequences. The methylotroph sequences represented three major families, Methylococcaceae, Methylocystaceae and Methylophilaceae. Within each family, a variety of phylotypes were detected suggesting complex community composition within each class. While the Methylococcaceae sequences were most numerous in each microcosm (50 to 100% of the methylotroph 16S rRNA sequences), including the unamended sample, shifts in phylotype composition occurred in response to different incubation conditions. The +O2+NO3- microcosm was characterized by relatively low diversity of Methylococcaceae, with sequences closely related to those of the characterized Methylobacter species being most numerous (Table 3 and Supplemental Table 1). The diversity of Methylophilaceae and Methylocystaceae was also low in this microcosm. Community diversity in the +O2-NO3-microcosm was somewhat higher, and most of the Methylococcaceae sequences were novel sequences that could not be affiliated with any described Methylococcaceae. The Methylophilaceae sequences were also most diverse in this microcosm, including unclassified Methylophilaceae. In the -O2-NO3- microcosm, only two methylotroph sequences were detected, both Methylococcaceae, while both Methylococcaceae and Methylophilaceae were detected in the -O2+NO3- microcosm.

Table 3 Distribution of 16S rRNA gene sequences among different genera (% of total methylotroph sequences distributed among the three families).

Family/Genus	Unamended	+O2-NO3-	+O2+NO3-	-O2-NO3-	-O2+NO3-	
Methylococcaceae						
Methylobacter ( ≥ 97%)	21.0	12.0	43.0		28.0	
Methylobacter (95%)	21.0	2.0	14.0	50	18.0	
Methylosarcina	11.0				9.0	
Methylomicrobium				50		
Methylosoma		8.0				
Unclassified Methylococcaceae	26.0	26.0				
Methylocystaceae						
Methylocystis		4.0				
Methylosinus		4.0	29.0			
Methylophilaceae						
Methylotenera		12.0	14.0		9.0	
Methylovorus	11.0	10.0			9.0	
Methylobacillus	5.0	2.0			9.0	
Methylophilus		4.0				
Unclassified Methylophylaceae	5.0	16.0			18.0	

The diversity of methane oxidizing bacteria was further assessed by profiling genes encoding subunits of both particulate (pmo) and soluble (mmo) methane monooxygenase enzymes. Profiling of the pmoB genes (the largest and the less conserved of the pmo genes) revealed significant diversity at the genus level suggesting the presence of at least five identifiable genera within Methylococcaceae and two genera within Methylocystaceae (Table 4), with an exception of the -O2-NO3- microcosm in which no pmoB genes were detected. The Methylocystaceae sequences were only detected in the aerobic microcosms, in agreement with the 16S rRNA gene profiling data. Once again, shifts in relative presence of different phyla were noted suggesting community dynamics in response to variable oxygen tensions and nitrate presence. The lowest phylotype diversity was observed for the -O2+NO3- condition, where only two methanotroph phylotypes were detected, Methylobacter and Methylovulum. In the aerobic microcosms, along with identifiable phylotypes, novel phylotypes of Methylococaceae were present, making up a significant proportion of the population. In the -O2+NO3- microcosm as well as in the unamended sample, pmo sequences were also identified belonging to the NC10 phylum, but in each case these constituted a minor proportion of the population. Very few mmo genes were detected in the metagenomes (Supplemental Table 2), suggesting that most of the active methane oxidizers were devoid of soluble methane monooxygenase.

Table 4 PmoB* diversity. Sum in each column equals 100%.

Genus/Family	Unamended	+O2-NO3-	+O2+NO3-	-O2-NO3-	-O2+NO3-	
Methylobacter	32.0	11.5	33.0		45.0	
Methylovulum	19.0	19.0	7.0		45.0	
Methylomonas	10.0	5.5	23.0			
Methylomicrobium	7.0	1.0	10.0			
Methylocaldum		1.0				
Unclassified Methylococcaceae	29.0	37.5	20.0			
Methylocystis		14.0	7.0			
Methylosinus		5.5				
Unclassified Methylocystaceae		4.0				
NC10	3.0	1.0			10.0	
Notes.

* PmoB is the alpha subunit of the particulate methane monooxygenase.

Genes encoding formaldehyde activating enzymes (fae) were profiled to evaluate the communities possessing formaldehyde oxidation potential. As expected, the diversity of fae genes/proteins extended beyond the bona fide methylotroph species (Table 5) and included Planctomycetes, Archaea, Burkholderiales as well as unclassified bacteria whose ability to oxidize or assimilate C1 compounds is unknown. The highest diversity of Fae was observed in the unamended sample and in the +O2-NO3- microcosm. In the latter, however, the non-methylotroph sequences made up a very minor proportion of total sequences while in the former they made up 30.3% of total sequences. In the enriched microcosms, with the exception of the -O2+NO3- condition, the majority of sequences were classified as belonging to Methylococcaceae and Methylophilaceae. The Methylocystaceae sequences were only prominently present in the +O2-NO3- microcosms. Only in the -O2+NO3- microcosm, the bona fide methylotroph fae genes constituted a relatively minor fraction of the total, with the dominant type being the planctomycete type. While Planctomycetes typically encode Fae and other functions of the tetrahydromethanopterin-linked formaldehyde oxidation pathway, methylotrophy has not been demonstrated in these species, and their genomes lack recognizable genes for methane oxidation or methanol oxidation functions (Chistoserdova, 2011b). The Methylococcaceae were most prominently represented by the Methylobacter and Methylomonas-like sequences, while within Methylophilaceae, the Methylotenera sequences were most prominent. Overall, the community structure predictions as deduced from Fae analysis agreed with other analyses (16S rRNA gene, pmoB and mmoX genes and other methylotrophy genes, not shown).

Table 5 Fae* diversity. Sum in each column equals 100%.

Taxon	Unamended	+O2-NO3-	+O2+NO3-	-O2-NO3-	-O2+NO3-	
Methylococcaceae						
Methylobacter	27.0	10.8	21.0	100	7.7	
Methylomicrobium	1.6	2.0				
Methylomonas	8.0	11.0	16.0		7.7	
Unclassified Methylococcaceae	6.4	9.5				
Methylocystaceae						
Methylocystis		6.3				
Methylosinus	1.6	9.5	5.0			
Unclassified Methylocystaceae		2.0				
Methylophilaceae						
Methylotenera versatilis	4.8	23.0	34.0			
Methylotenera mobilis	1.6	10.8	13.0		15.4	
Methylovorus	11.3	9.0	8.0		7.7	
Methylobacillus	0.8	0.7				
Unclassified Methylophilaceae	6.5	1.3				
Other						
Burkholderiales	8.0	0.7			15.4	
Unclassified Proteobacteria	3.2	1.3				
Unclassified	1.6		3.0			
Planctomycetes	6.4	0.7			30.7	
NC10	1.6					
LW phylum	1.6	0.7			7.7	
Archaea	8.0	0.7			7.7	
Notes.

* Fae is formaldehyde activating enzyme.

Methanol is the primary product of methane oxidation, and thus methanol dehydrogenase is an essential enzyme in the methane oxidation pathway (Anthony, 1982). In methanotrophs, this reaction is carried out by an enzyme encoded by mxaFI genes (Chistoserdova & Lidstrom, 2013). mxaFI genes are also essential in methanol oxidation by other groups of methylotrophs (Chistoserdova & Lidstrom, 2013). However, recently methylotrophs capable of methanol oxidation were described not possessing mxaFI genes, and in these, either mdh2 or xoxF genes have been implicated in this function (Chistoserdova & Lidstrom, 2013). Previous metagenomic analysis of Lake Washington populations suggested that the abundant Methylophilaceae phylotypes lacked mxaFI genes but possessed multiple copies of xoxF (Kalyuzhnaya et al., 2008). Genomes of both alpha- and gammaproteobacterial methanotrophs are also known to encode xoxF genes (Chistoserdova, 2011b). The metagenomes generated in this study were specifically analyzed for the presence of mxaF and xoxF genes that classed in the families of Methylococcaceae, Methylocystaceae and Methylophilaceae, using high stringency BLAST searches (Supplemental Table 3). In each category, both types of genes were detected. As previously observed (Kalyuzhnaya et al., 2008), the Methylophilaceae populations appeared to be dominated by the types lacking mxaF. The Methyococcaceae xoxF genes also appeared to be more abundant compared to the mxaF genes with the exception of the +O2-NO3- microcosm. This may suggest either that some of the Methylococcaceae lack mxaF genes or that some of them contain multiple copies of xoxF genes.

The diversity of genes for denitrification and relative abundance of genes belonging to key methylotroph groups (the Methylococcaceae, the Methylophilaceae and the Methylocystaceae) was evaluated by taxonomic profiling of the genes identified by the IMG/M platform as nitrate reductase, nitrite reductase, nitric oxide reductase or nitrous oxide reductase genes. For comparison, diversity and relative abundance of methylotroph nitrogenase genes were evaluated, revealing a significant phylogenetic complexity of the communities with a potential for denitrification (Table 6 and Supplemental Tables 4–8). We found that in the aerobic microcosms, the nitrogenase, nitrate reductase and nitrite reductase genes taxonomically ascribed to Methylococcaceae and primarily to the genus Methylobacter, were most abundant (Fig. 4), suggesting that Methylobacter was the major species capable of both nitrogen fixation and denitrification in these conditions, as well as in the native sediment. Methylophilaceae-affiliated nitrate- and nitrite reductases represented another dominant class in the aerobic microcosms (Fig. 4). The most abundant nitric oxide reductase type in these microcosms was classified as Methylophilaceae and most prominently Methylotenera, suggesting that most of Methylotenera species encode this step of the pathway while some of the Methylococcaceae represented in the metagenomes lack the respective gene. The phylotypes classified as Methylobacter were notably under-represented in this category (Supplemental Table 6). No nitrous-oxide reductase genes were detected affiliated with either Methylococcaceae or Methylophilaceae suggesting that the denitrification pathway may be incomplete in these species. However, a number of sequences were classed with Methylocystaceae in the +O2-NO3- microcosm (Fig. 4). Analysis of the nitrogenase genes demonstrated that Methylococcaceae and most prominently Methylobacter species were the major type possessing the potential for nitrogen fixation (up to 80% of total sequences), with a prominent presence of Methylocystaceae in the +O2-NO3- microcosm (Fig. 4, Supplemental Table 8). The remaining sequences of the denitrification and nitrogen fixation genes were distributed evenly among a variety of phyla, and no other dominant groups or groups specifically responding to nitrate were detected (Supplemental Tables 4–8).

Table 6 Relative abundance and diversity of nitrate metabolism genes. Phylotypes are defined as unique taxon IDs assigned by BLAST to nr.

Microcosm/protein	Total phylotypes	Unamended	+O2-NO3-	+O2+NO3-	-O2-NO3-	-O2+NO3-	
Nitrate reductase	239	975a (0.37%)b 162c	697 (0.44%) 116	259 (0.57%) 59	51 (0.24%) 33	541 (0.29%) 135	
Nitrite reductase	207	454 (0.17%) 128	448 (0.28%) 94	167 (0.37%) 45	29 (0.13%) 24	225 (0.12%) 98	
Nitric oxide reductase	118	200 (0.08%) 59	235 (0.15%) 49	135 (0.30%) 31	19 (0.09%) 18	121 (0.07%) 57	
Nitrous oxide reductase	52	102 (0.04%) 32	75 (0.05%) 28	12 (0.026%) 6	15 (0.07%) 8	83 (0.05%) 25	
Nitrogenase	66	172 (0.07%) 39	353 (0.22%) 30	123 (0.27%) 17	6 (0.03%) 4	45 (0.02%) 19	
Notes.

a Total number of genes annotated.

b Percent of total annotated enzymes.

c Number of phylotypes.

Figure 4 Relative abundance of nitrate metabolism genes ascribed to Methylococcaceae (blue), Methylophilaceae (red) and Methylocystaceae (green). Other (purple) represents a variety of phylotypes, including methylotrophs of other families, present at low abundances. See Supplemental Tables 4–8 for statistics.

Discussion

The metagenomic approaches, including “functional metagenomics” allow glimpses into the content of natural microbial communities, including uncultivated species, along with understanding their most prominent activities in global elemental cycles (Chistoserdova, 2010; Morales & Holben, 2011). We have previously employed a “high-resolution” metagenomics approach to communities inhabiting freshwater sediment using stable isotope probing (SIP), in order to specifically target populations involved in utilization of single carbon compounds with a few notable outcomes (Kalyuzhnaya et al., 2008). In this previous work we uncovered a dominant presence of Methylobacter species as part of the bacterial community actively consuming methane in this environment, in contrast to the results from cultivated methanotroph species (Auman et al., 2000). We also discovered a prominent presence of novel Methylophilaceae species that were classed into a separate, novel genus, Methylotenera (Kalyuzhnaya et al., 2006). These species appeared to be active in consuming a variety of C1 substrates, most notably methylamine, methanol and methane (Kalyuzhnaya et al., 2008). As we were able to cultivate Methylotenera species at the same time (Kalyuzhnaya et al., 2006; Kalyuzhnaya et al., 2012), another contradiction arose: as expected for members of the Methylophilaceae (Anthony, 1982), these species contained no genes that would encode methane oxidation functions. No genes for a typical (MxaFI) methanol dehydrogenase were present in these organisms (Lapidus et al., 2011). How then could they successfully compete for carbon from either methane or methanol with other species that possess the traditional enzymes for such types of metabolism? One other notable discovery was the persistent presence of genes for the denitrification pathway in Methylotenera species, suggesting a potential connection between methylotrophy and denitrification and a potential for electron acceptor alternatives to oxygen (Kalyuzhnaya et al., 2008). However, in the laboratory the cultivated Methylotenera species revealed very low potential for methanol metabolism (Kalyuzhnaya et al., 2006; Kalyuzhnaya et al., 2012). However, further experiments with in situ populations using labeled methanol, varying tensions of oxygen and varying presence of nitrate have confirmed that the Methylophilaceae, and most prominently the Methylotenera species, must be the major methanol utilizers in Lake Washington (Kalyuzhnaya et al., 2009). XoxF, a homolog of the traditional methanol dehydrogenase (large subunit) was proposed as a gene involved in methanol oxidation (Beck et al., 2011), supported by high expression of these genes in in situ conditions (Kalyuzhnaya et al., 2010).

In this work, we pursued three major objectives: determining what, if any, guilds beyond Methylococcaceae and Methylocystaceae were involved in methane oxidation in freshwater lakes, whether Methylophilaceae were involved in this process, and if the presence of nitrate had an effect on methane-oxidizing communities. We demonstrate that the known methanotroph guilds, Methylococcaceae and Methylocystaceae appear to be the major responders to the methane stimulus in aerobic microcosm incubations. More specifically, the Methylococcaceae and species belonging to or related to the genus Methylobacter are both the dominant species in the natural environment as well as the dominant responders to methane and to nitrate in aerobic conditions. While sequences of the recently described methanotroph guild NC10 that carries out methane oxidation anaerobically and links it to nitrite or nitrate (Wu et al., 2011) were detected in all samples, these were minor members of the community, and no response to methane or nitrate was observed. Even though the abundance of Methylococcaceae sequences in microaerobic microcosms was much lower compared to both aerobic microcosms and to the unamended sample, they significantly outnumbered the NC10 sequences. No other phylum revealed a pattern suggesting involvement in methane oxidation, and no novel methane monooxygenase genes were detected, suggesting that in both aerobic and microaerobic conditions methane was metabolized by the methanotrophs traditionally called “aerobic methane oxidizers” (Chistoserdova & Lidstrom, 2013). Methanotrophs of the family Methylococcaceae revealed a pronounced positive response to the addition of nitrate in aerobic conditions. However, these organisms do not appear to encode a complete respiratory denitrification pathway and likely use nitrate and nitrite reductases for assimilating nitrogen. Most if not all of these organisms also encode nitrogenases. The Methylocystaceae that constitute a smaller population in Lake Washington sediment also positively responded to methane but not to nitrate, in aerobic conditions, but they were almost undetectable in microaerobic conditions. The only non-methanotroph guild that responded to methane and nitrate stimuli was the Methylophilaceae, of which Methylotenera species were the most prominent in the datasets analyzed. Moreover, the response pattern of the Methylophilaceae correlated well with the pattern of the Methylococcaceae in aerobic conditions, suggesting a potential cooperation between the two groups at ambient oxygen tension. On the contrary, at low oxygen tension, and especially in the presence of nitrate, high community diversity, including the diversity of the denitrification genes, was observed, suggesting cross-feeding from labeled metabolites originating from the methanotrophs, even though the latter were present at a low population level.

The nature of the cooperation between the Methylococcaceae and the Methylophilaceae is not obvious. It could be suggested that the methane oxidizers release methanol as a result of high activity of methane monooxygenase, and that the Methylophilaceae consume this methanol, quickly incorporating it into their biomass. However, the dominant population of the Methylophilaceae enriched in the methane-fed microcosms appears to be most closely related to Methylotenera versatilis, cultivated representatives of which grow poorly if at all on methanol and lack bona fide (MxaFI) methanol dehydrogenase (Kalyuzhnaya et al., 2012). On another hand, multiple guilds that are known to be robust methanol oxidizers, such as Methylobacteriaceae, Hyphomicrobiaceae, Xanthobacteriaceae, as well as methanol-oxidizing Methylophilaceae (Methylovorus, Methylophilus) are minor members of the enriched communities. The Methylophilaceae could be involved in detoxification of nitrogen species to some of which, most notably ammonia, Methylococcaceae are known to be sensitive (Nyerges & Stein, 2009). However, in this case it is difficult to explain why Methylophilaceae are more successful than other species active in nitrogen metabolism. The same argument would be appropriate if a non-specific cross-feeding (for example on metabolites resulting from lysis of the Methylococcaceae) is suggested. The analyses presented here suggest that methanotrophs known as “aerobic” methanotrophs appear to be responsible for metabolizing methane in both aerobic and microaerobic conditions, even though they appear not to be as efficient at low oxygen tension as they are at high oxygen tension. The Methylophilaceae appear to be involved in methane oxidation in the aerobic conditions but not in microaerobic conditions, suggesting that the methanotrophs, dependent on the specific environmental circumstances, may engage in different types of partnerships, involving either a very specialized guild such as methylotrophs of the family Methylophilavceae or a diverse group of heterotrophs with versatile metabolic repertoires.

Conclusions

The well-characterized “aerobic” methanotrophs and most prominently the Methylobacter species are responsible for metabolism of methane in Lake Washington sediment in both aerobic and microaerobic conditions. In aerobic conditions, some type of a cooperative behavior between the Methylococcaceae, and most prominently the Methylobacter species is suggested by our data with the Methylophilaceae species, among which the Methylotenera species are most prominent. The nature of this type of cooperation remains unknown and requires a separate investigation. Both functional groups respond positively to the addition of nitrate. However, their ability to carry out classic respiratory denitrification is unlikely, as is a direct metabolic linkage between methane oxidation and denitrification. It is more likely that nitrate stimulates the methylotroph communities as a nutrient.

Supplemental Information

Supplemental Figure 1 Non-metric multidimensional scaling (NMDS) of Bray–Curtis dissimilarity matrix.

Click here for additional data file.

Supplemental Tables 2 and 3 Relative abundance of mmoX genes, Relative abundance of MxaF/XoxF genes.

Click here for additional data file.

Supplemental Tables 4--8 Distribution of 16S rRNA genes among major phyla, classified to the family level.

Click here for additional data file.

Supplemental Table 1 Counts of genes for nitrogen metabolism.

Click here for additional data file.

Additional Information and Declarations

Competing Interests

Author Contributions

DNA Deposition

L Chistoserdova is an Academic Editor for PeerJ.

David A.C. Beck analyzed the data, wrote the paper.

Marina G. Kalyuzhnaya conceived and designed the experiments, performed the experiments, wrote the paper.

Stephanie Malfatti analyzed the data.

Susannah G. Tringe and Tijana Glavina del Rio contributed reagents/materials/analysis tools.

Natalia Ivanova analyzed the data, contributed reagents/materials/analysis tools.

Mary E. Lidstrom conceived and designed the experiments, wrote the paper.

Ludmila Chistoserdova conceived and designed the experiments, analyzed the data, wrote the paper.

The following information was supplied regarding the deposition of DNA sequences:

IMG/M database, http://img.jgi.doe.gov/cgi-bin/m/main.cgi

IMG submission IDs 1377, 1378, 1873, 1879, 1888.

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
