# Peer review of "A metagenomic insight into freshwater methane-utilizing communities and evidence for cooperation between the Methylococcaceae and the Methylophilaceae"

_PeerJ, doi:10.7717/peerj.23_

## Round 0.1 · original submission · Minor Revisions

This paper describes bacterial communities that can use methane derived from sediment in Lake Washington and examined in the context of 4 microcosms. The authors provide evidence for potential cooperation between two methane using bacteria in a coordinated response to methane and nitrate. They have conducted careful experiments and obtained a nice modern data set, using 13C-methane in 4 microcosms with different conditions and both 16S and shotgun WGS of all 4 microcosms in addition to the sediment itself. The use of SIP isolates the organisms involved in the process they are interested in, a very effective and still somewhat unusual approach. While their excel-generated description of the alpha diversity does enable some aspects of the data to be described, this is too much data for the eye to compare entirely in terms of abundance in tables and pie charts. The microbial ecology community has developed user friendly tools for this type of analysis which would greatly enhance the readers ability to understand the conclusions presented here, and may also help the authors understand their data better and most of all, help the reader.

Reviewer 1 ·

Basic reporting

No Comments

Experimental design

No Comments

Validity of the findings

No Comments

Additional comments

Major

The figures do not capture some of the most interesting aspects of their data, including the 13C methane experiment timeline and the beta diversity comparing the microcosms, differences between 16S and WGS metagenome sequence data. I suggest the use of MG-Rast, mother and/or Qiime for the 16S data, ideally followed by ecology ordination analysis such as MDS using a Bray-Curtis distance matrix from the abundance tables (this can be done in R, see the vegan package and others, Primer, Qiime and many other software).

Throughout the text, when the taxonomy of hits to a particular gene are shown, they are referred to as a phylogenetic profile. My understanding of a phylogenetic profile is of an analysis of the patterns of co-occurance of multiple proteins across phyla, which sometimes helps t annotate unknown proteins that may be part of the same pathway. A phylogenetic profile of the sequenced methane-utlizing genomes described in this manuscript and compared with what is found in the metagenomes does sound interesting. The approach taken by Eisen and colleagues using freely available software designed for microarray analysis might be useful (see Fig 7 from PLoS Genet. 2005 Nov;1(5):e65. Epub 2005 Nov 25.)

Otherwise, please replace the term "phylogenetic profile" with "taxonomy of" when describing the taxonomy of genes found in the metagenomes.


Suggestions for figures:

- Consider replacing Pie charts with bar or lattice charts (google "Why pie charts are bad")
- add a figure with a schematic describing the microcosms and and experimental design

- A figure with days to heavy band on the X axes and microcosm on the Y axes
- Figure 1 in the form of bar charts and with the addition of MDS or unifrac or another clustering technique would be much easier to understand
- MDS or other analysis comparing pyrotag and WGS sequencing


Minor

- The figures need labels in the figure itself rather than only in the legend. Rather than giving details about those legends, re-structuring as described above should hopefully lead to legends that contain a single important message rather than catalog-like descriptions of large amounts of data.

- Labelling sections in the text with headings describing what is in the section would also help guide the reader

- Figure 3: it would be helpful to understand the overall abundance of what we are looking at here. In 15-20, are there few reads attributed here overall, as they are mostly composed of elements that are described as rare overall? A legend (labels for the colors in the figure itself) would also be very helpful here. The first line of the figure legend here and for all figures should summarize an important result from the figure rather than just describing an encyclopedic catalog

line 105: genes annotated as such > identified as

line 110: do you mean all genes from those organisms, or just the sequence for pmoB and mmoX?

line 210: being matched > matching
line 212 respectively) the > respectively). The
line 276 missing period after Figures 1,2)

Table 1: add read length and # of reads
many table legends: remind readers what function the abbreviated gene names encode

·

Basic reporting

no comments

Experimental design

no comments

Validity of the findings

no comments

Additional comments

This is a well executed and well written study. Minor suggestions for improvement are:
1) watch for consistent spelling of genus names throughout
2) since there was a such a major difference in the number of reads between libraries, the authors might want to comment on how the data were normalized in order to make proper comparisons in percentages of certain gene categories between experiments.
3) in the conclusion, mention which groups of methanotrophs were likely cooperating with the methylotrophs
4) since nitrate had a strong effect on the populations, are the authors suggesting that it is stimulating the community as an N-source? If so, then perhaps instead of ending the paper with the conclusion that the population is likely not carrying out respiratory denitrification, they could conclude with the possibility that nitrate is stimulating growth as a nutrient.

·

Basic reporting

I enjoyed reading Beck et al. Paper. The authors wanted to show three things: Determining guilds involved in methane oxidation in freshwater lakes; 2. The role of Methylophilaceae in methane oxidation and; 3. The effect of nitrate on methane-oxidizing communities. They sampled a 63 m depth sample of sediment and used WGS metagenomics and amplicons of 16S rRNA for species diversity profiles, as well as some genes involved in methane oxidation as phylogenetic markers. They perform microcosm experiments enriching the atmosfere with isotope labeled 13CH4 and playing with different concentrations of Nitrogen and Oxigen to perform the experiments and measure the communities changes across time of incubation under this conditions.

The paper contributes with understanding on methane oxidating species diversity and its changes across the time, assuming that the isotope marked 13C is incorporated into its DNA. However the experimental design can not help to distinguish the atmosphere enrichment from the natural death/decay of the community members across time. So the paper needs to be rephrased to make clear this point. And specifically to remove the statement that the methane-oxidizing species are product of the enrichment, that could be, but the experimental design doesn't help to conclude this. This concern is detailed in Experimental Design section of the report.

Think that the authors' expertise will answer/solve the particular observations in a short time and would be enough to publish the article.

Experimental design

My main concern about the design is that the authors are enriching the microcosm atmosphere (4 treatments) with very different times of incubation (10, 15, 20, and 30 days), and then use time 0 to compare the community structure. The concern here is that not being able to differentiate between the effect of the atmosphere enrichment or the aging/dead of the least resilient members of the community. I would suggest for further studies to have controls for each one of the incubation times, unamended samples of the same age for each incubation time to differ between treatments.

Validity of the findings

The main findings of the paper will remain, but discussion and conclusions need to be softened due to the experimental design. The authors asses the community structure by different approaches and this is solid. Although some remarks on how to select the thresholds is required and needs to be detailed in the methods section of the paper.

Additional comments

For the nitrogen metabolism genes relying on the annotation (p5 L122-123) only of the NCBI's (nt) makes me think about the huge amount of possible false positives lying on database due to annotation errors spreading. Would suggest to use a curated database and then search through profiles or position algorithms like hmmer or psi-blast to precisely annotate your dataset.

P6 L137-156 This section should fit better into methods of the paper, not results.

P7 L159-162. Please move this paragraph into methods.

P7 L163;171 The clusters are made at a 97% sequence identity, this is also known as OTUs, in this results you are only showing the clustering numbers, but neither here or in any table I looked into founded the total amount of sequences without clustering. This helps to see if you have some bias due to uneven coverage of each of the samples. Please incorporate the raw pyrotag sequences numbers. It would be helpful too to use an estimator of species abundance that does not rely on sequencing depth like the non-parametric estimators CHAO1 or ACE.

P7 L 171;173 This is one of the phrases not supported by the design: pleas rephrase from increased, decreased from one condition into other.

Even though the authors count with 5 WGS metagenomes the soul of the paper relies on phylogenetic profiling most of it with single gene approaches. The genes are obtained via PCR and then pyrotag sequencing or taxonomically assigned through an identity percentage. I think it would help to the reader to include precisely the rational behind the thresholds used in this study (i.e. p5. L114-116). P8 L199-201 Please include references to the thresholds here. Are you using Konstantinidis & Tiedje cut-offs?

Goris, J., Konstantinidis, K. T., Klappenbach, J. a, Coenye, T., Vandamme, P., & Tiedje, J. M. (2007). DNA-DNA hybridization values and their relationship to whole-genome sequence similarities. International journal of systematic and evolutionary microbiology, 57(Pt 1), 81–91.

Konstantinidis, K. T., & Tiedje, J. M. (2005). Genomic insights that advance the species definition for prokaryotes. Proceedings of the National Academy of Sciences of the United States of America, 102(7), 2567–72.

Have you considered to use whole genome recruitment of the metagenomes? I think it could help a lot to the paper, with figures to each reference genome, and think would not be time consuming like here:

Belda-Ferre P, Cabrera-Rubio R, Moya A, Mira A (2011) Mining Virulence Genes Using Metagenomics. PLoS ONE 6(10)

p10, L 233 consider use bin rather than categorized

p12, L287 what is a significant sequence sampling?

p12, L298 were most numerous. How most numerous than? Please indicate the raw numbers.

p12, L303 also low. The whole sentence does not make sense without numbers.

P12, L306 most diverse.

p.13, L314 significant diversity = ? Please try to get diversity indexes (Simpson, Shannon, CHAO1, ACE).

p. 15, L384 most abundant = ?

Through out the text, be careful with the use of response, because it is not possible to say that this is a direct effect of the enrichments.

Table 1. Add the raw count of pyrotag sequences, and at least one diversity index across the samples.

Table 2. How is it possible to have more sequences assigned at the 90% than at the 60% cut-off? See row 1 of Methylobacter, Methylocystis, and Methylotenera

Table 4, 5, I would suggest to use taxonomy profile/distribution rather than diversity.

Table 6 And total number of OTUs?

Figure 1. Some tags in the figure would help a lot to understand, just a bracket for A,B that states Pyrotags. A second bracket for C, D that states metagenomes

Figure 3. Please change the numbers from X-axis use symbols rather than numbers, and use a key to identify each treatment. * + % # - or whatever sign you choose rather than numbers from 1 – 5 will help the reader a lot to get into the figure.

---

## Round 0.2 · accepted · Accept

Thanks very much for promply making all the suggested corrections, I think the paper is really interesting and should be published.